# Optimal Protein Intake in Healthy Children and Adolescents: Evaluating Current Evidence

**DOI:** 10.3390/nu15071683

**Published:** 2023-03-30

**Authors:** Maria Garcia-Iborra, Esther Castanys-Munoz, Elena Oliveros, Maria Ramirez

**Affiliations:** 1Abbott Nutrition R & D, Granada University Science Park, 18016 Granada, Spain; maria.garcia16@abbott.com (M.G.-I.); esther.castanysmunoz@abbott.com (E.C.-M.); 2Abbott Nutrition R & D, Abbott Laboratories, 18004 Granada, Spain; elena.oliveros@abbott.com

**Keywords:** protein intake, children, adolescent, body composition, BMI, FMI, FFMI

## Abstract

High protein intake might elicit beneficial or detrimental effects, depending on life stages and populations. While high protein intake in elder individuals can promote beneficial health effects, elevated protein intakes in infancy are discouraged, since they have been associated with obesity risks later in life. However, in children and adolescents (4–18 years), there is a scarcity of data assessing the effects of high protein intake later in life, despite protein intake being usually two- to three-fold higher than the recommendations in developed countries. This narrative review aimed to revise the available evidence on the long-term effects of protein intake in children and adolescents aged 4–18 years. Additionally, it discusses emerging techniques to assess protein metabolism in children, which suggest a need to reevaluate current recommendations. While the optimal range is yet to be firmly established, available evidence suggests a link between high protein intake and increased Body Mass Index (BMI), which might be driven by an increase in Fat-Free Mass Index (FFMI), as opposed to Fat Mass Index (FMI).

## 1. Introduction

Nutrition plays an essential role in the health, function, and proper development of children and adolescents [1]. In particular, nutrition is a key element for disease prevention, especially for chronic diseases such as obesity and Type II Diabetes Mellitus (T2DM) [2,3,4]. The two first years of life have been identified as a critical window that may influence health later in life [3,5,6]. High protein intake in infancy has been linked to increased adipogenesis and a higher risk of obesity across one’s lifespan [3,6]. In response to this concern, regulatory bodies decreased the protein levels in infant formula (IF) in an effort to better mimic human milk for infants who cannot be breastfed [7,8,9]. Currently, the permitted minimum and maximum protein content in both infant and Follow-On Formulas (FOF) are 1.80 g and 2.50 g per 100 kcal, respectively, according to the Commission Delegated Regulation (EU) 2016/127 [10], which are below the maximum levels of the previous regulation (2006/141/EC) (3 g/100 kcal).

While, in infants, a lower protein may be favorable, there are other life stages and populations where increased protein intakes are recommended to promote beneficial health effects [11,12,13]. According to the Academy of Nutrition and Dietetics (Academy), Dietitians of Canada (DC), and the American College of Sports Medicine (ACSM), physically active individuals should consume between 1.2 and 2.0 g/kg body weight per day to support metabolic adaptation, repair, remodeling, and protein turnover. This range, compared with the Recommended Dietary Allowance (RDA) for non-pediatric populations (0.8 g/kg bw/d), supposes an increase of approximately 0.4–1.2 g/kg bw/d [14,15]. Additionally, higher intakes may be indicated for short periods during intensified training, or when reducing energy intake [14]. Similarly, in order to minimize the adverse health consequences of the reduced muscle mass observed with ageing, experts suggest that increasing protein intake above the recommended levels, to about 1.2 to 2.0 g/kg bw per day or higher, may be an effective nutritional strategy for elderly adults [11,16]. An increased protein intake for this population, which may improve muscle health, prevent sarcopenia, maintain energy balance, assist with weight management, and support cardiovascular function [11,16], represents an approach to maintain independence and improve the quality life in healthy elderly adults.

Despite the emerging evidence, as well as attempts to elucidate optimal protein intake levels for specific populations, there is a scarcity of data available assessing optimal protein intake in pediatric populations older than infants (<1 years) and toddlers (1–3 years), according to the age group classification of the European Food Safety Authority (EFSA) [17]. Hörnell et al. [18] reviewed scientific data on the short- and long-term health effects of different levels of protein intake from 0 to 18 years of age, identifying a total of 23 studies that assessed the impact on growth and body composition. Nevertheless, only six of them (four prospective cohorts and two interventional studies) evaluated protein intake beyond 4 years and its impact later in life, and none assessed protein intake beyond 10 years of age. However, it is worth mentioning that the interventional studies could be defined as short-term, since, in both cases, the supplementation was for only 7 days [19,20].

Once current recommended protein intake is achieved in children and adolescents aged 4 to 18 y, increased intakes could be beneficial to promote proper development, or, on the contrary, they may lead to increased risk of diseases later in life. Some critical time windows for protein intake have been hypothesized, such as during the transition to a family diet [21]. This period is usually characterized by a rapid increase of protein intake, largely due to the shift to cow’s milk, which has a protein content of approximately 5.15 g/100 kcal, about two times higher than that of IF or FOF [22].

Protein intake trends in children and adolescents in Western Europe and United States (US) are usually two- to three-fold higher than the dietary recommendations [21,23,24,25]. However, these recommendations are calculated by the factorial method using data from nitrogen balance studies [26,27]. Nitrogen balance weaknesses have been thoroughly discussed in the literature, as they tend to overestimate nitrogen intake and underestimate excretion, resulting in a net positive balance, and potentially underestimating one’s actual requirements [28,29]. This raises the question of whether current protein recommendations are accurate, or underestimate the actual needs of children and adolescents. Further research is needed to identify possible critical time windows, determine optimal protein intake ranges, and investigate whether current recommendations should be reevaluated.

To address these concerns, this narrative review has been structured into four sections. The first section provides an overview of current protein intake trends in developed countries and compares them with dietary reference intakes for each age group. Given that protein consumption is likely to exceed recommendations, the next section consists of a discussion on the accuracy of current protein recommendations, as well as new techniques for assessing protein metabolism in children. The third section presents a detailed examination of available evidence regarding protein intake in children and adolescents aged 4 to 18 years, and its effects later in life. To organize the findings, studies have been categorized into three sections based on their primary outcomes, including body mass index, body composition outcomes, and insulin sensitivity. Lastly, a short section to explore the role of protein in picky eaters, a complex behavior characterized by food refusal, has been included.

## 2. Current Protein Intake Trends

Current evidence suggests that the majority of Western Europe’s and the United States’ population exceeds dietary protein recommendations [15,26,30]. Recommendations for protein intake in the pediatric population (<18 years old) vary slightly depending on the issuing health authority, as shown in Table 1. The most common referenced recommendations are the Dietary Reference Intakes (DRIs) from the US Institute of Medicine (IoM) [30] and the Dietary Reference Values (DRVs), published by the EFSA [26]. While these recommendations encompass different terms, the RDA and the Population Reference Intake (PRI) are the most cited ones, since they refer to the average daily dietary intake level that is sufficient to meet nutrient requirements and prevent deficiencies in practically all (97.5%) the healthy individuals in a population [15,31]. However, beyond the dietary recommendations to prevent deficiency, there are no guidelines for an “optimal” protein intake in pediatric population for promoting healthy growth and development.

In particular, in developed countries, children’s and adolescent’s protein intake trends are usually two- to three-fold higher than the recommendations, and these proteins are mostly derived from animal sources [21,23,24,25]. Data from 2016/2017 to 2018/2019 outlined National Diet and Nutrition Survey (NDNS) [25] (n ≈ 1000; 500 adults and 500 children), conducted in the United Kingdom (UK), concluded that the mean protein intake was 52.9 and 64.5 g/d in the ranges 4–10 years and 11–18 years, respectively. Regarding PRI for these ages, protein consumption was 2.39 and 1.40-fold higher, respectively. In other European countries, protein intake is even higher. The National Dietary Survey on the Child and Adolescent Population in Spain (ENALIA) [24], based on 1862 participants, found that average intakes ranged from 74.45 (4–8 years) to 93.6 g/d (14–17 years), with 3.88–1.88-fold higher values than PRI, respectively. It should be noted that for children aged 4–17 years, protein’s contribution to total energy consumption was up to 17.8%. Similar results have been found in US. The National Health and Nutrition Examination Survey 2001–2014 (NHANES) [23] (n = 15.829; aged 2 to 80 years) showed average intakes, ranging from 59.7 g/d, in children aged 4–8 years, to 79.75 g/d, in those 14–18 years. It is worth mentioning that 0.96% of children aged 2–3 years had values above the specific Acceptable Macronutrient Distribution Range (AMDR) for protein (10–30 %E), while, in 1–3 years Spanish children, this value was 12.1%. AMDR expresses a range of protein intakes in the context of a complete diet; nevertheless, in adults, the upper limit of AMRD (35 %E) has been associated with a risk for prediabetes and T2DM [11,15,32]. As a result, AMDR should be considered with caution. Lastly, in agreement with these national dietary surveys, original studies extracted from cohorts, such as the Dortmund Nutritional and Anthropometric Longitudinal Design (DONALD) study [21] or Generation R [33], reported similar, higher protein intakes.

In conclusion, while observational studies have consistently shown that the average protein intake among children is two- to three-fold higher than the recommended dietary intakes, which aim to prevent deficiency in 97.5% of the population, there are currently no guidelines for an “optimal” protein intake that promotes healthy growth and development beyond these recommendations.

## 3. Dietary Protein Intake Recommendations: A Discussion

In adults, nutrient requirements are defined as the minimum continuous daily intake needed to prevent deficiencies [15]. In children, however, this concept acquires greater importance, as nutrients should support normal growth and development [34]. The recommended protein intake for children and adolescents is derived by the factorial method, using data from nitrogen balance studies, and considers requirements for both growth and maintenance [26,27]. However, the nitrogen balance technique has several methodological and data interpretation drawbacks in determining protein recommendations, which have been thoroughly discussed elsewhere [29,35]. Briefly, this technique tends to overestimate nitrogen intake and underestimate excretion, resulting in an overly net positive balance, and, therefore, this leads to an underestimation of the requirements. Moreover, the measurement process requires several days of adaptation to the protein intake level used for testing, and additional time is needed for the corresponding measurements. In addition, this process must be repeated at a minimum of three protein intake levels to assess the zero balance [28,29]. Furthermore, data interpretation limitations have been extensively examined, resulting in an independent reanalysis of nitrogen balance studies to determine alternative reference values for adults [28]. Nevertheless, this technique continues to be the “gold standard” for determining protein requirements [26,30].

New techniques to assess protein metabolism in children are emerging, such as Indicator Amino Acid Oxidation (IAAO), ^15^N End-Product, and D_3_-Creatine methods [29]. These three techniques consist of minimally invasive methods for study participants in a free-living environment, and can be applied in vulnerable populations, such as children and adolescents. Specifically, the IAAO method had already been studied in children aged 6–10 years by Elango et al. [36], and its results are likely to be less prone to error than those from nitrogen balance studies [28,29,36,37].

Elango et al. [36] suggest that current protein requirements in healthy school-aged children (6–10 years old) are severely underestimated. Protein requirements were determined to be 1.3 and 1.55 g/kg bw/d for Estimated Average Requirements (EAR) and RDA respectively; these values are significantly higher than the DRI values for protein (71.0% and 63.2% higher, respectively). Additionally, the authors reported similar conclusions, in adults, in two different ways: the IAAO method, and by reanalysis of the current nitrogen balance data by using nonlinear regression [28,37].

Additionally, Gattas et al. [38] conducted a study to determine protein requirements in 8–10 year-old healthy children, by using nitrogen balance data. The results obtained were 0.94 g/kg bw/d for EAR and 1.2 g/kg bw/d for RDA. These results represent higher values than those obtained for DRIs, even when they were obtained by linear regression. If those results are reanalyzed by using a more appropriate method, such as two-phase linear regression analysis, the results obtained are 1.13 and 1.44 g/kg bw/d for EAR and RDA, respectively. Hence, these new values suggest a possible underestimation of DRIs and EFSA recommendations for protein intake in children, as shown in Table 2.

Lastly, it should be mentioned that certain life factors may alter optimal protein needs in children and adolescents. One of the most reported factors is physical activity, which has not been considered in calculating DRIs, despite it having great relevance in child development [39,40]. Current evidence suggests that physical activity may increase protein requirements due to enhanced muscle protein synthesis and breakdown [34,41,42,43,44,45,46,47]. Nevertheless, there is not enough published evidence to determine whether alternative protein recommendations are needed, or by how much protein requirements may need to be increased, in children with heightened physical activity. Emerging protein determination methods represent a great step forward to assess the life factors that influence protein intake and constitute a much-needed area of research [29].

Based on this evidence, emerging techniques to assess protein metabolism in the pediatric population highlight the need to reevaluate current recommendations. These recommendations not only appear to be underestimated, but also fail to consider various life factors that can affect optimal protein needs in children.

## 4. Long-Term Effects of Protein Intake in Healthy Children and Adolescents

With the aim to explore the possible long-term effects of protein intake in healthy children and adolescents, we evaluated available studies and extracted their data in Table 3. A consensus on the definition of high protein intake has not been reached, thus the included studies defined it as either the highest quantile of protein intake among the evaluated cohort, or as the mean intake above the recommendations for the population studied. Because of this lack of standardization, together with the heterogeneity observed among studies, the evidence has been thoroughly discussed to understand the long-term effects of high protein intake in this population. Additionally, to ensure accurate results, only studies that considered potential confounders, such as total energy intake, non-energy-providing nutrients, and early life and socioeconomic factors, have been included.

Of the 14 studies included in this review, 13 are observational, while only 1 is a randomized controlled trial investigating the effects of dairy protein on children’s growth and body composition [48]. The thirteen observational studies reported results from eight cohorts, with sample sizes ranging from 70 to 3991. Cohorts were predominantly from Europe and US, with the DONALD study being the most referred to cohort [21,49,50,51]. Protein intake was assessed as grams per day or percentage of total energy intake (%E).

### 4.1. Long-Term Effects of Protein Intake on Body Mass Index

Several obesity outcomes were evaluated in the studies included, with body mass index (BMI) or body mass index standard deviation score (BMI-SDS) being the most widely reported (n = 8). From the thirteen observational studies included in this review, seven assessed two or more obesity measures [21,33,50,52,53,54,55]. The most common outcomes, after BMI or BMI-SDS, were the Fat Mass Index (FMI) and Fat-Free Mass Index (FFMI), or the lean mass index (n = 5) [33,50,53,54,55]. The remaining two studies examined BMI-SDS together with body fat percentage (BF%) [21], and BMI-SDS together with triceps (TR) and subscapular skinfolds (SS) measurements [52].

Eight observational studies reported the relationship between protein intake and BMI (n = 3) [33,56,57] or BMI-SDS (n = 5) [21,49,52,53,58] from seven cohorts. Of them, six studies reported positive associations [21,33,49,53,57,58], one reported a negative association [52], and one reported a non- significant association [56]. Moreover, three studies examined associations with animal protein intake [21,49,53], all of them reporting a positive association with a higher BMI later in life.

Günther et al. [21], in 203 individuals of the DONALD study, examined whether high protein intake at five different time points (6 m, 12 m, 18–24 m, 3–4 y, and 5–6 y) was related to increased BMI-SDS or BF% at 7 years. Results suggest two critical periods, 12 months and 5–6 years, in which higher animal protein intakes were positively associated with BMI-SDS (12 m: *p* for trend = 0.02; 5–6 y: *p* for trend = 0.07) and BF% at 7 years (12 m: *p* for trend = 0.008; 5–6 y: *p* for trend = 0.01). However, results for protein intake at 5–6 years were overall less convincing than those for 12 months. Additionally, it is worth mentioning that a higher protein intake from dairy products at 12 months was associated with a higher BMI-SDS later (*p* for trend = 0.02); meanwhile, there was no relationship at 5–6 years (*p* for trend = 0.7). In this line, a higher vegetable protein intake at 5–6 years (*p* for trend = 0.05) (but not at 12 months) showed an inverse relationship with BF% later in life. Hence, results suggest there may exist different effects on body composition, depending on the protein source and the age at which the consumption takes place.

Analyzing a bigger sub-cohort of the DONALD study (1028 healthy children and adolescents, aged 2 to 18 years), Hermanussen [49] showed a significant association between BMI-SDS and total and animal protein intake, both when expressed as mean absolute daily protein intake and as a percentage of daily energy intake. In particular, the correlation depended on age and reached a maximum in the group of 10–12 years, where protein intake explained up to 13% of the BMI variance. Corroborating this evidence, a longitudinal study developed by Skinner et al. [57] in 70 children aged 2–8 years suggests that mean longitudinal dietary protein intake during this age range (boys ≈ 56.89 ± 15.11 g/d; girls ≈ 53.44 ± 16.33 g/d) was positively associated with BMI at 8 years.

The association between protein intake and BMI has also been examined among 1999 children enrolled in the Generation XXI cohort [58]. They demonstrated that higher protein intake at 4 years of age was positively associated with BMI-SDS at 7 years in both boys and girls. Nevertheless, it is worth highlighting that even the lowest tertile of protein intake, which was considered as the reference category, represents 5.19- and 4.98-fold higher intake when compared to the EFSA recommendations, for boys and girls, respectively [26]. In contrast, an Italian study with similar mean protein intakes [56] reported no significant association between protein intake at 8 years (73 ± 16 g/d) and body composition impact, expressed as BMI, four years later. Regarding PRI values from EFSA, they were about 2.92-fold increased.

In the light of this evidence, high protein intake may have different impacts on body composition depending on age, which is also supported by additional studies [21,33,59]. Nevertheless, it should be highlighted that, although mean protein intake (g/d) was similar in both studies, when expressed as %E, it was above 15 %E (boys: 18.6 ± 2.58 (%E); girls: 18.8 ± 2.54 (%E)) only in the study conducted by Durao et al. [58]. Although these values are in accordance with AMDR (10–30 %E), in some cases, an upper limit of this dietary reference has been associated with increased risk of later obesity in infants [18], and with prediabetes and T2DM in adults [32]. Whether this is the case in the 4–18 year-old age group remains to be determined.

Magarey et al. [52] also explored the relationship between macronutrient intake and BMI, TR, and SS from 2 to 15 years of age. For most ages, energy-adjusted macronutrient intakes at previous ages were not significant predictors of BMI-SDS at subsequent ages. Exclusive protein intake at 6 years (boys = 54.7 ± 13.3 g/d; girls = 51.1 ± 12.3 g/d) was negatively associated with the body composition outcome at 8 y. Regarding TR and SS measurements, protein intake was not a significant predictor of TR-SDS or SS-SDS, for any time interval.

Considering these findings as a whole, an apparent positive trend relationship was found between protein intake and BMI. However, these results must be evaluated cautiously, in order to accurately understand the implications for pediatric health. It should be noted that BMI alone is considered to be of limited use when examining body composition in children. There is increasing evidence that BMI is a poor body fat mass indicator, and it does not reflect the location of fat mass [60,61]. Thereby, it would be more appropriate to measure fat and fat-free mass, or lean mass. This is of great significance, since it has been suggested that protein’s influence in mid-childhood may be mainly explained by FFMI, and not FMI [33]. Indeed, lean mass is an essential component, as it plays an important role in maintaining posture and normal movement in children, as well as in adolescents [62].

**Table 3 nutrients-15-01683-t003:** Characteristics of 14 studies describing long-term effects of protein intake in healthy children and adolescents.

Observational Studies										
Source	Age (Years)	No. of Individuals	% of Boys	Age of Outcomes	Total Protein Intake	Animal Protein Intake	Dairy Protein Intake	Plant Protein Intake	Main Outcomes	Observations
Günther et al., 2007 [21], the DONALD Study, Germany	1; 5–6	203	50.25	7	1 y: 13.3 (11.7, 14.8) (%E)	1 y: 8.4 (7.1, 9.8) (%E)	1 y: 4.4 (2.5, 6.3) (%E)	1 y: 4.8 (4.0, 5.7) (%E)	BMI-SDS at 7 y: positive association; a higher animal (*p* for trend = 0.03) and dairy (*p* for trend = 0.02) protein intake (%E) was associated with a higher BMI-SDS at 7 y. BF% at 7 y: positive association; a higher animal (*p* for trend = 0.008) and dairy (*p* for trend = 0.07) protein intake (%E) was associated with a higher BMI-SDS at 7 y.	In the fully adjusted models, the results for protein intake at 5–6 years were overall less convincing than those for 1 years.
					5–6 y: 12.4 (11.2, 13.7) (%E)	5–6 y: 7.8 (6.7, 8.8) (%E)	5–6 y: 3.5 (2.6, 4.5) (%E)	5–6 y: 4.5 (4.0, 5.1) (%E)	BMI-SDS at 7 y: positive tendency; a higher animal protein intake (%E) (*p* for trend = 0.07) was associated with a higher BMI-SDS at 7 y. BF% at 7 y: positive association; a higher animal protein intake (%E) (*p* for trend = 0.01) was associated with a higher BF% at 7 y. Inverse association: a higher vegetable protein intake (%E) *(p* for trend = 0.05) was associated with a lower BF% at 7 y.	
Magarey et al., 2001 [52], South Australia	2; 4; 6; 8; 11; 13; 15	143–243	51.1–59.0	4; 6; 8; 11; 13; 15	6 y boys: 54.7 ± 12.3 (g/d) 14.0 ± 2.1 (%E) 6 y girls: 51.1 ± 12.3 (g/d) 14.6 ± 2.0 (%E)	NA	NA	NA	BMI-SDS at 8 y: negative association; only protein intake at 6 years was negatively associated with BMI-SDS score at 8 y. Triceps measurement: no association; neither protein intake nor any nutrient was a significant predictor of the triceps-SD score for any time interval.	For most ages, energy-adjusted macronutrient intakes at previous age were not significant predictors of BMI-SDS at subsequent ages. None of the nutrients at 2 years were a significant predictor of BMI at 8 years.
Hermanussen 2008 [49], the DONALD Study, Germany	2–18	1028	48.7	--------	NA	NA	NA	NA	BMI-SDS: positive association; significant interaction with the mean absolute daily intake of all protein (r = 0.143, *p* < 0.0001) and animal protein (r = 0.151, *p* < 0.0001). Significant interaction with protein intake (%E) of all protein (r = 0.203, *p* < 0.0001) and animal protein (r = 0.163, *p* < 0.0001).	The correlation depended on age: maxima in the group of 10–12 years in both genders (boys: r = 0.31, *p* < 0.0001; girls: r = 0.36, *p* < 0.0001).
Skinner et al., 2004 [57], United States	2–8	70	52.86	8	Boys: 56.89 ± 15.11 (g/d) ^1^ Girls: 53.44 ± 16.33 (g/d) ^1^ Both: 14 (%E)	NA	NA	NA	BMI at 8 y: positive association; mean longitudinal (2–8 years of age) dietary protein intake (g/d) was positively related to BMI.	
Switkowski et al., 2019 [53], Project Viva Cohort (NCT02820402), US	3.2	1165	50	7.7 y 13 y	Boys: 58.2 ± 8.20 (g/d) Girls: 58.4 ± 8.48 (g/d)	NA	NA	NA	BMI z-scores at 13 y: positive association; a 10 g/d increase in total protein intake at 3.2 years was associated with 0.12 (95% CI: 0.01, 0.23) unit greater BMI z-scores only in boys. Positive association with animal protein intake only in boys (13 y). DXA lean mass index at 13 y: positive association; there was a trend towards a higher DXA lean mass index (*p* = 0.06) only in boys (13 y). Positive association with animal protein intake only in boys (13 y). Free IGF-I concentrations at 13 y: positive association; only in boys did a 10 g/d increase in total protein intake at 3.2 years correspond to a 5.67% higher total IGF-I (95% CI: 0.30%, 11.3%) and a 6.10% higher free IGF-I (95% CI: 1.19%, 11.3%).	Outcomes evaluated: height, IGF-I, measures of adiposity and lean mass. There were no associations of protein intake in early childhood with any of the mild-childhood (7.7 years) and adolescent girls (13 years) outcomes. There were no associations of protein intake in early childhood with either SS + TR skinfolds or DXA fat mass among boys at 13 years.
Durao et al., 2017 [58], Generation XXI, Portugal	4	1999	51.3	7	Boys: 77.5 ± 16.00 (g/d) 18.6 ± 2.58 (%E) Girls: 73.8 ± 14.33 (g/d) 18.8 ± 2.54 (%E)	NA	NA	NA	BMI z-scores at 7 y: positive association; higher protein intake in both boys (T3 vs. T1: *p* for trend = 0.045) and girls (T2 vs. T1: *p* for trend = 0.266) was positively associated with BMI z-scores. FSI at 7 y: positive association; higher protein intake was positively associated with FSI only in boys (T3 vs. T1: *p* for trend = 0.035). When compared to boys in the first tertile, boys in the highest tertile of protein intake at 4 years of age showed a statistically significant increase in FSI of 0.207 z-score units, at 7 years.	Boys: T1 (≤72.7 g/d) T2 (72.8–81.0 g/d) T3 (≥81.0 g/d) Girls: T1 (≤69.7 g/d) T2 (69.8–77.5 g/d) T3 (≥77.5 g/d)
van Vught et al., 2010 [55], CoSCIS, Denmark	6	203	46.31	9	Boys: 69.90 ± 17.33 (g/d) Girls: 62.72 ± 14.92 (g/d) Total: 66.05 ± 16.04 (g/d)	NA	NA	NA	Height at 9 y: positive association; high intake of ARG (*p* = 0.05) was associated with increased height among girls. Intake of protein or LYS was not associated with changes in linear growth either in boys or girls. FMI at 9 y: inverse association; high protein (*p* = 0.01), ARG (*p* = 0.01), and LYS (*p* = 0.01) intake was associated with a decrease in body fat gain in girls with a BMI in the 5th percentile. Inverse association between the intake of LYS and change in FMI, only among boys with a BMI in the 5th percentile (*p* = 0.01). FFMI at 9 y: no significant associations were found between habitual protein intake, ARG or LYS intake, and change in FFMI either in boys or girls.	Boys: ARG intake: 2.8 ± 0.9 (g/d) LYS intake: 4.5 ± 1.3 (g/d) Girls: ARG intake: 2.4 ± 0.6 (g/d) LYS intake: 4.0 ± 1.1 (g/d)
Jen et al., 2018 [33], Generation R Study, The Netherlands	8	3991	49.3	10	61.4 ± 17.1 (g/d) 16.5 ± 2.3 (%E)	38.1 ± 14.1 (g/d)	NA	23.3 ± 7.1 (g/d)	BMI at 10 y: positive association; a higher protein intake was associated with a higher BMI (model 3: 0.05 SDS, 95% CI 0.01, 0.09). Weight at 10 y: positive association; a higher protein intake was associated with a higher weight (model 3: 0.08 SDS, 95% CI 0.02, 0.13). Association mainly explained by a higher FFMI (model 3: 0.07 SDS per 5 %E, 95% CI 0.02, 0.11) and not FMI (model 3: 0.03 SDS, 95% CI −0.01, 0.07).	Both plant and animal proteins were associated with a higher FFMI, but the association was stronger for plant protein (model 3: 0.11 SDS, 95% CI 0.02, 0.21) They observed a trend between higher plant protein intake and lower FMI, which was significant when it was consumed at the expense of animal protein.
Maffeis et al., 1998 [56], Italy	8	112	51.79	12	73 ± 16 (g/d) 14.7 ± 1.7 (%E)	NA	NA	NA	BMI at 12 y: not significant association with protein intake (%E) (*p* = NS).	
van Vught et al., 2009 [54], EYHS, Denmark	9	364	44.27	14–16	Boys: 71.7 ± 21.4 (g/d) Girls: 67.8 ± 19.0 (g/d) Total: 70 ± 21 (g/d)	NA	NA	NA	FMI at 14–16 y: inverse association; high protein (*p* = 0.03), ARG (*p* = 0.04), and LYS (*p* = 0.03) intake was associated with a decrease in body fat gain only in girls with a BMI in the 1–4th (leaner girls). FFMI at 14–16 y: positive association; high protein intake was positively associated with an increase in FFMI gain only among girls with a BMI in the 5th quintile (*p* = 0.04). Boys: no significant associations were found for protein or for ARG or LYS and FMI or FFMI.	Boys: ARG intake: 3.0 ± 1.2 (g/d) LYS intake: 4.4 ± 1.8 (g/d) Girls: ARG intake: 2.8 ± 1.0 (g/d) LYS intake: 4.1 ± 1.5 (g/d)
Assmann et al., 2013 [50], the DONALD Study, Germany	Boys: 10–15 Girls: 9–14	262	46.56	18–25	Boys: T1 12.0 (11.3, 12.6) (%E) T2 13.3 (12.7, 13.7) (%E) T3 14.6 (14.0, 15.4) (%E) Girls: T1 11.2 (10.9, 11.8) (%E) T2 12.8 (12.2, 13.4) (%E) T3 14.4 (13.4, 15.2) (%E)	Boys: T1 7.0 (6.6, 7.5) (%E) T2 8.4 (8.1, 8.8) (%E) T3 10.0 (9.5, 10.6) (%E) Girls: T1 6.4 (5.9, 6.6) (%E) T2 7.9 (7.7, 8.2) (%E) T3 9.6 (9.0, 10.4) (%E)	Boys: T1 3.4 (2.5, 4.3) (%E) T2 4.2 (3.7, 4.7) (%E) T3 4.8 (3.8, 5.8) (%E) Girls: T1 3.4 (3.0, 3.9) (%E) T2 3.8 (3.1, 4.5) (%E) T3 4.6 (3.6, 5.3) (%E)	Boys: T1 5.1 (4.6, 5.5) (%E) T2 4.8 (4.2, 5.2) (%E) T3 4.6 (4.1, 4.9) (%E) Girls: T1 5.0 (4.5, 5.5) (%E) T2 4.8 (4.2, 5.3) (%E) T3 4.6 (4.2, 5.1) (%E)	FMI at 18–25 y: inverse association; a higher animal protein intake during puberty was associated with a lower FMI only in men (*p* for trend = 0.001). FFMI at 18–25 y: positive association; a higher animal protein intake was associated with a higher FFMI, primarily among women (*p* for trend = 0.001). Slightly positive association with a higher animal protein in young adult men (*p* for trend = 0.04).	There was no significant relationship between dairy protein intake during puberty and FFMI in young adulthood (*p* for trend = 0.17). Plant protein was not associated with body composition among either sex. Even in the highest energy-adjusted tertile of animal protein intake, protein accounted for less than 15% of energy intake.
Joslowski et al., 2013 [51], the DONALD Study, Germany	Boys: 10–15 Girls: 9–14	213	44.6	18–36	Boys: T1 11.8 ± 1.1 (%E) T2 13.2 ± 0.8 (%E) T3 14.5 ± 0.9 (%E) Girls: T1 11.2 ± 0.9 (%E) T2 12.8 ± 0.8 (%E) T3 14.5 ± 1.2 (%E)	Boys: T1 7.0 ± 0.9 (%E) 38.4 (g/d) T2 8.3 ± 0.4 (%E) 42.9 (g/d) T3 9.8 ± 0.9 (%E) 49.8 (g/d) Girls: T1 6.2 ± 0.9 (%E) 26.2 (g/d) T2 7.9 ± 0.4 (%E) 32.8 (g/d) T3 9.8 ± 1.0 (%E) 41.6 (g/d)	Boys: T1 3.5 ± 1.2 (%E) T2 4.4 ± 0.7 (%E) T3 4.6 ± 1.3 (%E) Girls: T1 3.5 ± 0.9 (%E) T2 3.7 ± 1.1 (%E) T3 4.6 ± 1.4 (%E)	Boys: T1 4.9 ± 0.8 (%E) T2 4.8 ± 0.8 (%E) T3 4.6 ± 0.7 (%E) Girls: T1 5.0 ± 0.7 (%E) T2 4.9 ± 0.7 (%E) T3 4.7 ± 0.7 (%E)	IGF-I and IGFBP-3 at 18–36 y: positive association; habitually higher animal protein intakes in females during puberty were related to higher IGF-I (*p* for trend = 0.005) and IGFBP-3 (*p* for trend = 0.01). IGFBP-2 at 18–36 y: inverse association; habitually higher animal protein intakes in females during puberty were related to lower IGFBP-2 (*p* for trend = 0.04).	Animal protein intake in puberty was not related to IGF-I, IGFBP-3, IGFBP-1, or IGFBP-2 in males. In contrast, among males, a habitually higher animal protein intake in early life (0.5–2 years) was associated with lower concentrations of IGF-I in young adulthood. Among females, animal protein intake in early life was not related to IGF-I. Data suggests that, among females, a habitually higher animal protein intake during puberty may precipitate an upregulation of the GH–IGF-I axis. By contrast, higher animal protein intakes in early life may yield a long-term downregulation of the GH–IGF-I axis in males.
Koppes et al., 2009 [63], AGAHLS, The Netherlands	13	350	48	36	NA	NA	NA	NA	Body fatness at 36 y: positive association; women with high body fatness at the age of 36 years had a significantly higher relative protein intake at ages 13 (*p* < 0.001), 32 (*p* < 0.05) and 36 years (*p* < 0.05). Men with high body fatness at the age of 36 years had a significantly higher relative protein intake at ages 32 (*p* < 0.05) and 36 years (*p* < 0.01).	Inter-period Pearson correlation coefficients are used to express the relative contribution to total energy intake of the four macronutrients. Throughout the 23-year period of follow-up, the relative protein intake in women with high body fatness at the age of 36 years was about 1% higher than in women without high body fatness.
**Interventional Studies**										
**Source**	**Study** **Design**	**Age** **(Years)**	**Sample Size**	**% of** **Boys**	**Intervention**	**Duration of** **Intervention**	**Baseline Total Protein** **Intake**	**Final Total Protein** **Intake**	**Main Outcomes**	**Observations**
Thams et al., 2022 [48], NCT03956732	2 × 2-factorial randomized controlled trial	6–8	184	45.11	Substitution of 260 g/d milk or yogurt in their diet with: (1) High-protein (HP) yogurt: 10 g protein/100 g (2) Normal-protein (NP) yogurt: 3.5 g protein/100 g	24 weeks (range: 21–26 weeks)	15.4 ± 2.4 (%E)	HP: 18.3 ± 3.4 (%E) NP: 15.9 ± 2.5 (%E)	The yogurt intervention per se resulted in a lower FMI increase with HP than with NP (*p* = 0.037). Regression analyses showed a negative dose–response association between changes in dairy protein intake (g/kg body weight/d) and changes in FMI (β: –0.19; 95% CI: –0.33, −0.041 kg/cm^2^; *p* = 0.012).	The expected intake of protein from the yogurts was around 17 g/d higher in HP than NP varieties, corresponding to an ∼25% increase in total protein intake for Danish children.

Values are means ± SD or median (25th and 75th percentiles). ^1^ Mean of protein intake values at age 2.3 to 8 y. AGAHLS, Amsterdam Growth and Health Longitudinal Study; ARG, arginine; BF%, Body Fat Percentage; BMI, Body Mass Index; BMI-SDS, Body Mass Index-Standard Deviation Score; CLHNS, The Cebu Longitudinal Health and Nutrition Survey; CoSCIS, Copenhagen School Child Intervention Study; DONALD Study, Dortmund Nutritional and Anthropometrical Longitudinally Designed Study; EYHS, European Youth Heart Study; FSI, Fasting Serum Insulin; HP, High protein; IGFBP, Insulin Growth Factor-Binding Proteins; Insulin-Growth-Factor I; LYS, lysine; NA, Not Available; NP, Normal protein; NS, No Significance; Q, Quartile; SCCNG, Southwest China Childhood Nutrition and Growth; T, Tertile.

### 4.2. Long-Term Effects of Protein Intake on Body Composition Outcomes

Among observational studies that reported both fat mass and lean mass outcomes, all described a positive association between a high protein intake and an increased lean mass (n = 4) [33,50,53,54], except one, which reported no significant association [55]. In contrast, regarding fat mass outcomes, one study reported positive association [33], three reported inverse associations [50,54,55], and one reported no associations [53].

Switkowski et al. [53], in a cohort of 1165 children from the U.S., examined associations between protein intake at 3.2 years and height, insulin-like growth factor type I (IGF-I), and measures of adiposity and lean mass at 7.7 years and 13.0 years. There were no associations between protein intake in early childhood and any of the mid-childhood outcomes. However, among adolescent boys, a 10 g/d increase in total protein intake at 3.2 years was associated with an increase in BMI-SDS by 0.12 units. Additionally, a trend towards a higher dual energy X-ray absorptiometry (DXA) lean mass index was associated with this. Nevertheless, there was no association of protein intake at 3.2 years with either DXA fat mass, SS or TR skinfolds among boys at 13 y. Similar results were found for animal protein intake, being positively associated with BMI-SDS and DXA lean mass in early adolescence among boys. It is worth mentioning that, despite absolute protein intakes being similar among adolescent boys (58.2 ± 8.20 g/d) and girls (58.4 ± 8.48 g/d), statistically significant associations were observed only in boys.

On the contrary, van Vught et al. [55], demonstrated that high protein intake may limit body fat gain over time, only among girls. Those associations took place when the girls were divided according to BMI quintiles. In accordance with this, the association between protein intake and a decrease in FMI was found among those girls with a BMI in the fifth quintile (mean protein intake: 59.7 ± 13.7 g/d). They evaluated arginine (ARG) and lysine (LYS) intakes and subsequent changes in FMI and FFMI, based on the theory that these amino acids may potently stimulate growth hormone (GH) secretion, and, therefore, may influence the development of body composition [64]. Results suggested that ARG and LYS, either given separately or in combination, were significantly related to a decrease in body fat gain among girls with a BMI in the fifth quintile. Regarding FFMI variance, no associations were found regarding high protein, ARG, or LYS intake, either in boys or girls. Furthermore, among girls, a high intake of ARG was associated with increases in linear growth.

The same researcher also demonstrated, only among girls, that high protein intakes at 9 years may significantly decrease body fat gain and increase fat-free mass gain at 14–16 years [54]. The association between protein intake and a decrease in FMI was found among leaner girls (BMI in the first to fourth quintile) (mean protein intake: 69.2 ± 18.8 g/d); meanwhile, FFMI was positively related only in girls with a BMI in the fifth quintile (mean protein intake: 62.3 ± 20.0 g/d). However, in both cases, an improvement in the body composition was found with an increased protein intake, compared to EFSA PRI (approx. 2.22- to 2.47-fold increases). The results suggested that ARG and LYS intakes were associated with a decrease in body fat gain only in leaner girls. On the other hand, among girls in the fifth BMI quintile, ARG intake was positively associated with FFMI, specifically when LYS intake was high. It has been suggested that the somatotopic regulation of growth, throughout GH and IGF-I production, could be modulated by specific amino acids [64,65]. Thereby, protein intake’s influence on body composition may not only be due to the total amount or source of the protein, but to the combination of specific amino acids.

These results suggest gender differences may exist, in terms of protein intake and subsequent body composition outcomes. This evidence is also supported by Assmann et al. [50] in a study conducted among 262 pubertal males and females that also studied outcomes in young adulthood (18–25 y). They found that a higher animal protein intake during puberty was associated with a lower FMI only in men. In contrast, a higher animal protein intake was associated with an increased FFMI, primarily among women. Additionally, a slightly positive association was found between higher animal protein intake and FFMI in young adult men. However, it should be noted that even in the highest tertile of both total and animal protein intake, protein accounted for less than 15 E%.

The effects of dietary macronutrient intake and body fatness on subjects, from the age of 13 years onwards, were also explored by Koppes et al. [63] over a period of 23 years. Regarding protein’s role, women with high body fatness at the age of 36 years had a significantly higher relative protein intake at ages 13, 32, and 36 years. Throughout the 23 years of follow-up, the relative protein intake in women with high body fatness at the age of 36 years was about 1% higher than in women without high body fatness. In contrast, among men with high body fatness, there were no higher relative protein intakes during puberty compared to men without high body fatness. As a study weakness, it should be noted that they did not evaluate if lean mass changed over this period.

Jen et al. [33] observed no gender differences in the associations between protein intake and body composition among 3991 children enrolled in the Generation R Study (Netherlands). They evaluated the effects of dietary protein intake at 8 years, and body composition up to age 10 years. Results concluded that a 5 %E higher protein intake was related to both higher weight and BMI, and was associated with a higher combined risk of overweightness and obesity up to 10 years, independent of whether it replaced carbohydrates or fat. This association is mainly explained by a higher FFMI and not FMI. It is worth mentioning that the mean protein intake was higher than 15 %E (16.5 ± 2.3 %E). Furthermore, both plant and animal protein were associated with a higher FFMI, but the association was stronger for plant protein. A trend between higher plant protein intake and lower FMI was observed, which was significant when it was consumed at the expense of animal protein. Therefore, this study supports that protein from plant sources may be beneficial for body composition. Accordingly, Günther et al. reported an inverse association between higher vegetable protein intake at 5–6 years and the BF% at 7 years.

As displayed in Table 4, observational studies assessing both fat mass and lean mass outcomes reported protein intake values far above PRI recommendations [33,50,53,54,55]. Specifically, data displayed fold-increased values ranging from 1.7 to 4.49, which translates into percentage increases between 70 and 348.46, compared to PRI values. Assuming that current protein requirements might be underestimated and novel values reported by Elango et al. may be more accurate, current protein intake would still exceed those recommendations [36]. However, evidence suggests that increases in protein intake between 70% and150%, compared to PRI values for prepubertal and pubertal children (8–15 years old), might positively correlate with FFMI and negatively with FMI, which implies an improvement in terms of body composition. Likewise, van Vught et al. [55], reported that body fat gains may be relatively prevented over time by high protein intake (3.48-fold-increase than PRI values), among 6 year-old girls.

Supporting this evidence, results of a randomized controlled trial suggested that an expected increase of 17 g/d of dairy protein for 24 weeks in 6–8 year-old children resulted in a significantly lower FMI increase [48]. To investigate the effects of high dairy protein intake on body composition, 184 children substituted 260 g/d dairy into their diet, with either high-protein (HP; 10 g protein/100 g) or normal-protein (NP; 3.5 g protein/100 g) yogurt, for 24 weeks. At baseline, total protein intake was 15.4 ± 2.5 %E; these values were similar, 15.9 ± 2.5 %E, for normal-protein yogurt intake and increased to 18.3 ± 3.4 %E with high-protein yogurt intake. The changes in FMI were consistent and dose-dependent, which may suggest causality. Therefore, despite FFMI being unaffected, a high protein intervention counteracted the increase in FMI in healthy 6–8 year-old children [48].

Similar results were also reported in an exploratory randomized controlled trial of just 4 weeks [66]. The effects of milk and rapeseed protein in 129 healthy 7–8 year-old children were evaluated by the intake of 35 g milk protein or 35 g milk and rapeseed protein (ratio 54:46 or 30:70) daily. At week four, dietary protein intake reached approximately 18 %E depending on the group studied (Milk group: 18.3 ± 1.9 %E; Blend protein 54:46 group: 18.4 ± 2.0 %E; Blend protein 30:70 group: 18.6 ± 1.9 %E). Despite the absence of a control group with a lower dose of protein intake, results from baseline to week four showed an improvement in body composition parameters. Specifically, BMI increased in all groups (*p* < 0.05), mainly due to an increase in FFMI (*p* < 0.01). Additionally, results suggested that FFMI increments were higher for milk alone than for rapeseed blends [66].

Taking these results together, an apparent positive trend exists between a higher protein intake and an increase in FFMI, but not FMI. The consistency of these observations appears to increase when protein consumption occurs during prepubertal and pubertal periods, and when the impact is assessed from adolescence onwards. This evidence also supports that high protein intake may have a differential impact on body composition depending on age. However, evidence remains unclear, and further research is needed to identify underlying gender and age differences regarding the associations between protein intake and body composition.

### 4.3. Long-Term Effects of Protein Intake on Insulin Sensitivity

With regards to protein intake and insulin sensitivity, three observational studies from the twelve included ones examined the association between dietary protein intake and later impacts on IGF-I, its binding proteins (IGFBP) and fasting serum insulin (FSI) [51,53,58]. All of them reported a positive association with high protein intake.

Physiologically, IGF-I levels and the expression of its receptors differ throughout life, according to which it may have a differential impact on body composition depending on age [67,68]. During the first year of life, its levels increase, showing a positive correlation with weight gain [68]. As a result, high IGF-I levels in infancy have been associated with later obesity [4,69]. Throughout childhood, IGF-I concentrations increase slowly, predicting height velocity in the following year. During puberty, IGF-I reaches its maximal levels, approximately at the age of 14.5 and 15.5 years in girls and boys, respectively, and gradually declines until the third decade [68,70,71]. Longitudinal studies have demonstrated that IGF-I is correlated with height velocity only in prepubertal children, despite its levels remaining elevated during puberty. In contrast, IGFBP-1 levels decrease during puberty, whereas IGFBP-2 levels remain unchanged throughout this period [68]. According to these variations, during normal puberty and adolescence, there is a decline in insulin sensitivity [71,72].

Durao et al. [58] observed a statistically significant positive association between higher protein intake at 4 years and FSI at 7 years. When compared to boys in the first tertile (≤72.7 g/d), boys in the highest tertile (≥81.0 g/d) of protein intake at 4 years showed a significant increase, of 0.207 z-score units, in FSI at 7 y.

On the contrary, in a cohort of 1165 American children [53], a positive association has been found between protein intake at 3.2 years and both total IGF-I and free IGF-I at 13 years, but not at 7.7. Particularly, a 10 g/d increase in total protein intake in early childhood corresponds to 5.67% higher total IGF-I and a 6.10% higher free IGF-I in boys aged 13 years, but no associations were found among adolescent girls. In terms of development, IGF-I has been demonstrated to be a robust predictor of bone mass in early puberty in both genders [73,74], an effect mainly mediated by increments in lean mass [75]. Supporting this evidence, Switkowski et al. also reported a positive trend between both total and animal protein intake and a higher lean mass index at 13 years in boys. Based on the theory of the “bone bank”, in which early deposits lay the foundation for skeletal health, it is worth mentioning that more than half of peak bone mass is acquired during the teen years [76]. Thereby, promoting proper bone mass development during childhood and adolescence could lead to a lower risk of bone diseases, such as osteoporosis, later in life. Similarly, Jolowski et al. [51] provided epidemiological evidence showing a relationship between habitually higher animal protein intake during puberty and increased IGF-I and IGFBP-3 levels, but only among young adult females (18–36 y). Among this population, an inverse association with IGFBP-2 is reported. These gender differences may be due to the presence of higher testosterone levels in boys, which may override a potential effect of animal protein intake on IGF-I [51]. Secondly, it is known that girls have a higher degree of physiological insulin resistance during puberty, thus, they may be more vulnerable to dietary effects on the GH–IGF-I axis than males [77]. The authors suggest that there is an upregulation of the GH-dependent components of the GH–IGF-I axis, which may be discernible in the long-term. In adults, IGF-I has been linked to an increased risk of breast cancer [78], but also to lower risk of cardiovascular disease [79], and osteoporosis [80], and has also been shown to be a protective factor against the development of glucose intolerance [81].

It is also important to note the study conducted by Hua et al. [82], in which prospective associations between the habitually higher protein intake of healthy children (3–17 y) and the impact on adult stature were assessed. Results suggest that total and animal protein intakes above dietary recommendations were prospectively, independently, and positively related to adult height only in girls. The authors propose that one major mechanism behind these associations would probably be elevated levels of growth factor activity, including as IGF-I. However, protein’s impact on IGF-I was not evaluated in this study.

Based on the evidence, some weaknesses can be noted. First, due to the observational nature of the included studies, causation cannot be inferred. Moreover, some studies evaluated BMI alone, which is considered to be of limited use to predict obesity. The fourteen included studies reported results from eight cohorts, highlighting that further investigations in other populations are warranted. Additionally, due to the lack of consensus on the definition of high protein intake and the wide range of ages and protein intakes studied, it becomes challenging to compare results and identify specific needs based on age. Lastly, limited studies explored the existence of a correlation between protein intake and body composition outcomes at various time points, hindering the determination of critical time windows for protein intake, as well as of the optimal protein intake range.

In conclusion, current evidence suggests an apparent positive trend between high protein intake and BMI. However, it may be mainly explained by an increase in FFMI and not FMI. Protein intake could modulate the GH–IGF-I axis, increasing IGF-I levels during puberty and young adolescence, which may promote proper bone and lean mass development, although this is largely influenced by the hormonal component. As a consequence, the relevance of gender differences increases during the pre-pubertal years. Hence, it seems that protein intakes higher than recommendations could have beneficial long-term effects on body composition. While the optimal range is yet to be firmly established, the literature advises caution regarding exceeding 15–20 %E in the early stages of childhood [18,21]. Further studies are warranted to explore the optimal protein intake, the role of different protein sources, even amino acids, and the underlying mechanisms that influence body composition later in life.

## 5. Optimal Protein Intake in Picky Eaters

Picky eating—also called ‘fussy’, ‘selective’, or ‘choosy’ eating—is a complex behavior, characterized by the consumption of an inadequate variety or quantity of foods through the rejection of a substantial amount of both familiar and unfamiliar foods [83,84,85]. However, as there is no universally accepted definition, it is complicated to identify picky eaters (PE), assess their incidence, and investigate the real consequences of such behavior. Despite this, several studies have addressed the subject in recent years [83,84,85].

Picky eating is usually associated with significantly lower intakes of fruits, vegetables, and meats [84,85,86,87,88]. Dairy intakes are generally similar between PE and non-PE [84,85], and so are energy and macronutrient intakes [84]. However, Samuel et al. [84] concluded that protein intakes were significantly lower in PE versus non-PE in five out of ten studies. When selecting only those aged 3 years old or above, five out of seven [87,88,89,90] studies showed significantly lower protein intake in PE. It may suggest that this behavior is accentuated with age, leading to a poor-quality diet. Accordingly, Taylor et al. [85] characterized, both cross-sectionally and longitudinally, the diets of children aged 10 and 13 y, suggesting that persistent PE (from 3 y) presented more pronounced differences at each age. Regarding protein intakes, the persistent PE group consumed 10% lower protein versus the non-PE group. In most studies [84,85], dietary protein intakes were nevertheless generally above EFSA dietary recommendations in all age groups.

There are scarce longitudinal studies focused on anthropometric characteristics and body composition in children who are picky eaters [91,92,93,94], which hinders drawing causal inferences. A recent analysis reported by Grulichova et al. [92] identified 346 participants as PE and 1722 as non-PE, among the cohort members of the European Longitudinal Study of Pregnancy and Childhood (ELSPAC-CZ). The adjusted models showed negative associations with weight (on average PE were about 2.3 kg lighter than non-PE) and height (on average PE were about 0.8 cm shorter than non-PE). These results are in line with those reported in the Avon Longitudinal Study of Parents and Children (ALSPAC) (7420 children) [93]. PE were identified at 3 years of age, while weight and height were measured at seven time-points between 7 and 17 years old. In this case, the models predicted that male and female persistent PE were about 1.5–2.5 kg and 1.0–1.5 kg lighter, respectively, compared to non-PE at each age. Results suggested a negative association with height, both in persistent PE boys (1.5–2.0 cm shorter) and girls (1.0–1.5 cm shorter). On the other hand, body composition was measured on five occasions at 9–17 years old, by DXA. Results suggested that persistent picky male children had a lower lean mass index compared to non-picky children at all ages, from 11 years onwards, by about 0.1 kg/m^2^, but there was no evidence of any differences in girls. Additionally, there was no association between picky eating and the percentage of body fat or FMI in either sex. Supporting this evidence, a study embedded in the Generation R prospective cohort (4191 children) suggested that young picky eaters (4 years) are at risk of having a lower fat-free mass and of becoming underweight at 6 years of age. Specifically, the picky eating profile was related to a lower BMI-SDS, mainly due to a decrease in FFMI, and not FMI [94].

In this population, the effects of oral nutritional supplements (ONS) have been explored as a means to improve nutritional status and promote growth. Four randomized controlled trials (RCTs) involved children with picky eating behaviors and intervened with ONS and dietary counseling (DC); this group was compared to another receiving DC alone [95,96,97,98]. Based on these RCTs, a metanalysis was performed, in order to show changes in growth parameters [99]. Regarding weight parameters, results suggest that the intervention group, in the four RCTs, shows significantly greater weight gain, weight-for-age z-scores, and weight-for-height z-scores at 30, 60, and 90 days compared with the group receiving DC alone. Conversely, three out of four RCTs reported a faster height gain in the ONS + DC group [99]. Specifically, Sheng et al. [95] showed no significant difference between the intervention and control group.

Other benefits derived from receiving ONS + DC were also reported. For instance, appetite significantly increased in the intervention group compared with the control group [96]. Ghosh et al. and Alarcon et al. suggest that the incidence of upper respiratory tract infections developed over the study period was significantly lower in the ONS + DC group compared to the control group [96,97]. Likewise, the effects of the high-protein ONS diet have also been assessed in healthy short and lean children [100,101,102]. Lebenthal et al. [100] evaluated the intake of a high-protein ONS (24.5 g protein/serving) diet for 6 months in 171 short and lean children aged 3–9 years. ‘Good’ formula consumers (intake of ≥ 50% of the recommended dose of one serving/day) significantly improved height-SDS and weight-SDS, with no change in BMI-SDS compared with ‘poor’ consumers and the placebo group [100]. In the follow-up study, Yackbovitch-Gavan et al. [101] reported similar results, suggesting that a one-year intervention with a high-protein ONS diet was effective in promoting the linear growth of these children, with no change in BMI. In contrast, in a similar approach in short and lean prepubertal boys (10–14.5 y), results reported a change in body composition [102]. The intervention included two phases: a double-blinded intervention with a high-protein ONS (36 g protein/serving) diet or placebo for 6 months, and subsequently, an open-label, extended 6-month diet containing ONS, for all the participants. It was reported that ‘good’ formula consumers showed significantly increased weight-SDS, BMI-SDS, fat-free mass and muscle mass when compared to ‘poor’ consumers and the placebo group [102]. Nevertheless, it should be mentioned that while the body composition described by the authors could be relevant to with PE children, the presence of picky eating or any other behavior related to the refusal to eat adequately was not assessed [100,101,102].

The available evidence shows some weaknesses. First, there is a scarcity of RCTs assessing ONS in PE, with only four studies found. The ONS consists of a balanced combination of macronutrients and micronutrients, with protein providing 12% to 15% total energy, so the health benefits could be owed to the set of nutrients and not just to the protein. Lastly, only changes in anthropometric parameters and not in body composition (FMI and FFMI) were assessed, so it does not allow discerning whether the changes are mainly explained by variations in the FFMI or the FMI.

Overall, despite PE children having apparently adequate protein intakes, lower fruit and vegetable intake and higher free sugar intake are markers of a poor diet [85,86,87,88]. The latest evidence seems to suggest that, on average, PE are shorter and have lower lean mass than their non-picky peers [91,92,93,94]. This implies that PE may benefit from early identification and intervention to promote proper growth and development. Whether they may benefit from additional protein intake as a nutritional intervention needs to be analyzed in well-designed RCTs.

## 6. Conclusions

In conclusion, our current narrative review provides a comprehensive overview of available evidence regarding protein intake in healthy children and adolescents aged 4 to 18 years, and its effects in later life. The main conclusions are:Observational studies have consistently reported that the average protein intake in children is two- to three-fold higher than the recommended dietary intakes. However, there are currently no guidelines for an “optimal” protein intake that promotes healthy growth and development in the pediatric population, beyond the recommended intakes to prevent deficiency.Emerging techniques to assess protein metabolism in children suggest current protein recommendations may be underestimated, implying a need for reevaluation.This review has not identified a relationship between a high protein intake in childhood and adolescence and obesity and insulin resistance later in life. The literature advises caution when exceeding 15–20 %E protein.Some beneficial effects may be associated with high protein intake in this stage of life, such as:
An apparent positive association between high protein intake and increased BMI, which was mainly explained by an increase in FFMI and not FMI.Protein intake could modulate the GH-IGF-I axis, increasing IGF-I levels during puberty and young adolescence, which may promote bone and lean mass development.In children with picky eating behaviors, a higher nutrient intake, including protein, is associated with positive changes in weight and height parameters.

## Figures and Tables

**Table 1 nutrients-15-01683-t001:** Current recommendations for protein requirements, estimated by age and sex, for children.

	EFSA ^1^			DRI ^2^			
	AR (g/kg bw/d)	PRI (g/kg bw/d)	PRI (g/d)	EAR (g/kg bw/d)	RDA (g/kg bw/d)	RDA (g/d)	AMDR (%E) ^3^
4–8 years	0.72	0.89	19.30	0.76	0.95	19	10–30%
9–13 years	0.72	0.90	34.50	0.76	0.95	34	10–30%
14–17 years, boys	0.71	0.88	53.25	0.73	0.85	52	10–30%
14–17 years, girls	0.69	0.85	46.50	0.71	0.85	46	10–30%

^1^ From Ref. [26]. ^2^ From Ref. [15]. ^3^ From Ref. [30]. AMDR, Acceptable Macronutrient Distribution Range; AR, Average Requirement; DRI, Dietary Reference Intakes; EAR, Estimated Average Requirements; EFSA, European Food Safety Authority; PRI, Population Reference Intake; RDA, Recommended Dietary Allowance.

**Table 2 nutrients-15-01683-t002:** Comparison of protein requirements in children 6–10 years old.

	Nitrogen Balance			IAAO ^3^
	DRI ^1^	Gattas et al. ^2^	Gattas et al. Reanalyzed ^3^	
EAR (g/kg bw/d)	0.76 (Ref)	0.94 (23.7% higher)	1.13 (48.7% higher)	1.3 (71.0% higher)
RDA (g/kg bw/d)	0.95 (Ref)	1.2 (26.3% higher)	1.44 (51.6% higher)	1.55 (63.2% higher)

^1^ From Ref. [15]. ^2^ From Ref. [38]. ^3^ From Ref. [36]. Adapted from Elango et al. [36]. DRI, Dietary Reference Intakes; EAR, Estimated Average Requirements; IAAO, Indicator Amino Acid Oxidation; RDA, Recommended Dietary Allowance.

**Table 4 nutrients-15-01683-t004:** Comparison of the increase in protein intake with respect to PRI and IAAO values.

Source	Age (Years)	No. of Individuals	Age of Outcomes (Years)	Total Protein Intake (g/d)	Total Protein Intake (%E)	PRI (g/d) ^1^	Fold Increase vs. PRI	% Increase vs. PRI	IAAO for 8.4 ± 1.4 Years (g/d) ^2^	Fold Increase vs. IAAO Results	% Increase vs. IAOO Results
Switkowski et al., 2019 [53], Project Viva Cohort (NCT02820402), US	3.2	1165	7.7 13	58.3 ± 8.34 ^3^	NA	13	4.49	348.46			
Van Vught et al., 2010 [55], CoSCIS, Denmark	6	203	9	66.1 ± 16.0 ^3^	13.91 ^3^	19	3.48	247.63	49.45 ^3^	1.34	33.58
Jen et al., 2018 [33], Generation R Study, The Netherlands	8	3991	10	61.4 ± 17.1	16.5 ± 2.3	25	2.46	145.6	49.45 ^3^	1.24	24.18
Van Vught et al., 2009 [54], EYHS, Denmark	9	364	14–16	70 ± 21	12.81 ± 3.84 ^3^	28	2.5	150	49.45 ^3^	1.42	41.57
Assmann et al., 2013 [50], the DONALD Study, Germany	Boys: 10–15 Girls: 9–14	262	18–25	68 ^3^	14.5 (13.7, 15.3) ^3^	40	1.7	70			

Values are means ± SD or median (25th and 75th percentiles). ^1^ From Ref. [26]. ^2^ From Ref. [36]. ^3^ Approximation calculated from data provided by original articles. CoSCIS, Copenhagen School Child Intervention Study; DONALD Study, Dortmund Nutritional and Anthropometrical Longitudinally Designed Study; EYHS, European Youth Heart Study; IAAO, Indicator Amino Acid Oxidation; NA, Not Available; PRI, Population Reference Intake; US, United States.

## Data Availability

Data sharing not applicable.

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
