# Peer review of "Optimal Protein Intake in Healthy Children and Adolescents: Evaluating Current Evidence"

_nutrients, 2023, doi:10.3390/nu15071683_

Round 1

Reviewer 1 Report

1. Sections running Line 109-131 - there is probably a need to add additional content at the start of this session to explain estimates of appropriate protein intake and link this section to the content of Table 1. The table above highlights values in g/kg body weight/d while the examples from different regions highlight values in g/d. Presumably each country/region has defined age-related cut-offs and come up with an estimated requirement in g/d based on an assumed body weight. I'm aware that examples of how this has been estimated are available, for example in the UK Government Dietary Recommendations.  (https://assets.publishing.service.gov.uk/government/uploads/system/uploads/attachment_data/file/618167/government_dietary_recommendations.pdf). I presume background information may also be accessible for some of the other countries/regions noted.

2. Table 2 - it may be useful to further consider these data to  define not just associations with BMI-SDS but to contextualise this in relation to findings (or lack of findings) on BMI status in children. Higher BMI-SDS within he underweight or ideal weight body weight status categories would be considered positive (perhaps positive associations with body fatness could only be considered positive in underweight individuals) in body fatness, while that would not be true in excess body weight categories. If the study populations were representative of European children, then I would assume overweight and obesity tended to be quite common among the cohorts/treatment groups? The relevance of any trends seen in observational data may need to be contextualised with the baseline body weight status of the population.

3. Line 305-306 - should this ready "may limit/reduce body fat gain" or similar to be more scientifically accurate?

4. Line 508 (and possibly elsewhere) - I recommend referring to "participants" rather than "subjects", as the latter term suggests individuals involved in the study did not give informed consent.

Author Response

Response to Reviewer 1 Comments Manuscript

Nutrients – 2273721

We are grateful for the reviewers’ comments. The manuscript has been modified according to the reviewers’ suggestions. We here below address the comments by the first reviewer. Changes in the manuscript are highlighted in track changes.

1) Point 1: Sections running Line 109-131 - there is probably a need to add additional content at the start of this section to explain estimates of appropriate protein intake and link this section to the content of Table 1. The table above highlights values in g/kg body weight/d while the examples from different regions highlight values in g/d. Presumably each country/region has defined age-related cut-offs and come up with an estimated requirement in g/d based on an assumed body weight. I'm aware that examples of how this has been estimated are available, for example in the UK Government Dietary Recommendations. (https://assets.publishing.service.gov.uk/government/uploads/system/uploads/attachment_data/file/618167/government_dietary_recommendations.pdf). I presume background information may also be accessible for some of the other countries/regions noted.

Response 1: We are grateful for the comments which would improve the clarity of the manuscript. We have addressed the suggestion of linking the mentioned section (Lines 109-131) by incorporating the Population Reference Intake (PRI) and Recommended Dietary Allowances (RDAs) into Table 1, expressed in grams per day (g/d) (EFSA NDA Panel 2012; Trumbo et al. 2002). PRI values are estimated based on the 50th percentile of reference body weights (kg) of European children (van Buuren, Schönbeck, and van Dommelen 2012). This approach enables us to avoid any discrepancies between country-specific requirements and provide better support for the original studies included in the narrative review, which mostly involve European cohorts.

Additionally, to address the variability in age ranges across different national surveys reported and align with the target population of the manuscript (4 to 18 years), we have provided an overview of the requirements estimates by age range in Table 1. This not only enhances clarity but also prevents potential confusion that may arise from presenting numerous data points based on the respective requirements of each national survey. Nevertheless, to ensure scientific accuracy, the individual comparisons for each national survey have been calculated using age-specific requirements outlined in the Scientific Opinion on Dietary Reference Values for protein published by the European Food Safety Authority (EFSA)(EFSA NDA Panel 2012).

2) Point 2: Table 2 - it may be useful to further consider these data to define not just associations with BMI-SDS but to contextualise this in relation to findings (or lack of findings) on BMI status in children. Higher BMI-SDS within he underweight or ideal weight body weight status categories would be considered positive (perhaps positive associations with body fatness could only be considered positive in underweight individuals) in body fatness, while that would not be true in excess body weight categories. If the study populations were representative of European children, then I would assume overweight and obesity tended to be quite common among the cohorts/treatment groups? The relevance of any trends seen in observational data may need to be contextualised with the baseline body weight status of the population.

Response 2: We thank the reviewer for the valuable input received. We believe the comment may refer to Table 3 rather than Table 2, as the latter does not present data on BMI-SDS. Table 3 presents the data reported by each study included, without interpretation. In our manuscript, we have used the term "positive association" in a statistical sense, which means that one variable tends to increase as the other variable increases. Conversely, a "negative association" indicates that one variable increases as the other decreases, and vice versa. By using these terms, we aim to convey the nature of the correlation between the two variables being studied, without interpreting whether the physiological impact of these associations is beneficial or not for health.

It is also important to note that although all the studies were conducted on healthy children living in developed countries, each study uses a different curve to calculate the BMI-SDS (Günther et al. 2007; Durao et al. 2017; Switkowski et al. 2019; Hermanussen 2008; Magarey et al. 2001). Moreover, some studies even standardize their data internally to adjust to their sample. Additionally, not all studies that evaluate the impact of protein intake on BMI-SDS report obesity or overweight, and those that do define it as “moderate” (Günther et al. 2007). Based on this, our aim is to report the evidence found by each study, comparing the impact to the baseline of its sample, without contextualizing it in the general population.

3) Point 3: Line 305-306 - should this ready "may limit/reduce body fat gain" or similar to be more scientifically accurate?

Response 3: We agree that using this language would be more accurate. We have updated the language accordingly.

4) Point 4: Line 508 (and possibly elsewhere) - I recommend referring to "participants" rather than "subjects", as the latter term suggests individuals involved in the study did not give informed consent.

Response 4: We thank the reviewer for the suggestion. We applied these changes to the manuscript.

References
Durao, C., A. Oliveira, A. C. Santos, M. Severo, A. Guerra, H. Barros, and C. Lopes. 2017. 'Protein intake and dietary glycemic load of 4-year-olds and association with adiposity and serum insulin at 7 years of age: sex-nutrient and nutrient-nutrient interactions', Int J Obes (Lond), 41: 533-41.

EFSA NDA Panel. 2012. 'Scientific Opinion on Dietary Reference Values for protein', EFSA Journal, 10: 2557.

Günther, A. L., T. Remer, A. Kroke, and A. E. Buyken. 2007. 'Early protein intake and later obesity risk: which protein sources at which time points throughout infancy and childhood are important for body mass index and body fat percentage at 7 y of age?', Am J Clin Nutr, 86: 1765-72.

Hermanussen, Michael. 2008. 'Nutritional protein intake is associated with body mass index in young adolescents', Georgian medical news, 156: 84-8.

Magarey, A. M., L. A. Daniels, T. J. Boulton, and R. A. Cockington. 2001. 'Does fat intake predict adiposity in healthy children and adolescents aged 2--15 y? A longitudinal analysis', Eur J Clin Nutr, 55: 471-81.

Switkowski, K. M., P. F. Jacques, A. Must, A. Fleisch, and E. Oken. 2019. 'Associations of protein intake in early childhood with body composition, height, and insulin-like growth factor I in mid-childhood and early adolescence', Am J Clin Nutr, 109: 1154-63.

Trumbo, Paula, Sandra Schlicker, Allison A. Yates, and Mary Poos. 2002. 'Dietary Reference Intakes for Energy, Carbohydrate, Fiber, Fat, Fatty Acids, Cholesterol, Protein and Amino Acids', Journal of the American Dietetic Association, 102: 1621-30.

van Buuren, S., Y. Schönbeck, and P. van Dommelen. 2012. "CT/EFSA/NDA/2010/01: Collection, collation and analysis of data in relation to reference heights and reference weights for female and male children and adolescents (0-18 years) in the EU, as well as in relation to the age of onset of puberty and the age at which different stages of puberty are reached in adolescents in the EU." In, 57. Draft Technical Report submitted to EFSA.

Reviewer 2 Report

It is a review article describing the optimal protein intake in Children and adolescents.

While it is important for Review article to provide a variety of content, it is also important to make it understandable to the reader. The authors summarized the main conclusion that the authors would like to mention in the conclusion section. However, it is too long, verbose and less readable. There is also a lack of summary or arrangement of key contents for each part. I think it will be necessary to organize the content a little overall.

line 25: As an example of chronic diseases that can be prevented through nutrition, it is better to present other chronic diseases rather than obesity.

Author Response

Response to Reviewer 2 Comments Manuscript

Nutrients – 2273721

We are grateful for the reviewers’ comments. The manuscript has been modified according to the reviewers’ suggestions. We here below address the comments by the second reviewer. Changes in the manuscript are highlighted in track changes.

1) Point 1: While it is important for Review article to provide a variety of content, it is also important to make it understandable to the reader. The authors summarized the main conclusion that the authors would like to mention in the conclusion section. However, it is too long, verbose and less readable. There is also a lack of summary or arrangement of key contents for each part. I think it will be necessary to organize the content a little overall.

Response 1: We are grateful for the valuable feedback provided by the reviewer, as it has allowed us to improve the clarity of our narrative review. We agree that it is crucial to ensure that our manuscript is understandable to the reader.

We conducted a narrative review to broaden our scope and evaluate not only the long-term effects of protein intake in healthy children and adolescents, but also to discuss protein intake in developed countries and provide insights into current protein requirements in the pediatric population, providing an overall contextualization of the situation. In addition, the considerable variability among the various studies assessing the long-term effects of protein intake, including differences in the studied population, age groups, and measured outcomes, poses a challenge to interpreting and summarizing the findings effectively. To address this concern, we categorized the content based on the primary outcomes measured by each study, resulting in three distinct sections, each with its respective arrangement of key contents. We have nevertheless taken note of the reviewer's suggestion and thoroughly revised the manuscript to address this issue. As a result, we have included an arrangement of the key conclusions for each section to facilitate understanding of the content. Additionally, we have improved the overall flow and readability of the text by revising the opening sentences of certain paragraphs to provide more detailed descriptions of their specific objectives. Furthermore, in Lines 83-95, we have provided a justification for the structure of the narrative review. Lastly, we have followed the reviewer's suggestion and condensed the conclusion section.

2) Point 2: Line 25: As an example of chronic diseases that can be prevented through nutrition, it is better to present other chronic diseases rather than obesity.

Response 2: As suggested by the reviewer, we have included type II diabetes mellitus as an additional example of chronic diseases that can be prevented through nutrition. It is worth mentioning that obesity represents a valuable example, as a growing body of evidence suggests that high protein intake during infancy increases the risk of obesity later in life (Rolland-Cachera et al. 1995; Camier et al. 2021). In fact, regulatory bodies recommended to lower protein levels in infant and follow-on formulas in response to this concern (Comission E). As it is well established that early protein intake is related to obesity, we wanted to address if this was also the case in a population from 4 to 18 years, given the scarcity of data available.

References
Camier, A., C. Davisse-Paturet, P. Scherdel, S. Lioret, B. Heude, M. A. Charles, and B. de Lauzon-Guillain. 2021. 'Early growth according to protein content of infant formula: Results from the EDEN and ELFE birth cohorts', Pediatr Obes: e12803.

Comission E. "Comission directive 2006/141/EC of 22 December 2006 on infant formulae and follow-on formulae and amending directive 1999/21/EC." In.

Rolland-Cachera, M. F., M. Deheeger, M. Akrout, and F. Bellisle. 1995. 'Influence of macronutrients on adiposity development: a follow up study of nutrition and growth from 10 months to 8 years of age', Int J Obes Relat Metab Disord, 19: 573-8.
